# AGILE: Development of a Compact, Low-Power, Low-Cost, and On-Board Detector for Ion Identification and Energy Measurement

**Florian Gautier on behalf of AGILE Collaboration**

Department of Physics and Astronomy, University of Kansas, Lawrence, KS 66045, USA;
florian.gautier.borrallo@ku.edu

**Abstract:** An AGILE (Advanced enerGetic Ion eLectron tElescope) instrument is being developed at the University of Kansas and NASA Goddard Space Flight Center to be launched on board a CubeSat in 2022. The AGILE instrument aims to identify a large variety of ions (H-Fe) in a wide energy range (1–100 MeV/nucl) in real-time using fast silicon detectors and fast read-out electronics. This can be achieved by the first use of real-time Pulse Shape Discrimination in space instrumentation. This method of discrimination relies on specific amplitude and time characteristics of the signals sampled every 100 ps and produced by ions that stop in the detector medium. AGILE will be able to observe, in situ, the fluxes of a large variety of particles in a wide energy range to advance our knowledge of the fundamental processes in the universe. This work presents the current stage of development of the instrument, the discrimination method used through the performed simulations, and the first results from lab tests using an Am-241 source.

**Keywords:** fast digital signal processing; particle measurements; silicon radiation detectors; space applications

## 1. Introduction

The development of robust particle detectors is a major interest of space research in order to probe the processes leading to the creation, transport, and loss of charged particles in the solar system [1]. Both the energy measurement and the identification of the particles species are a requirement for the development of such detectors. NASA has been greatly interested in the analysis of these particles and has studied them directly from space through missions, such as the SOHO mission [2], that focus on solar flares, but also by studying Earth's environment and magnetic field trapping those particles, such as the Van Allen probes mission [3], which discovered the constant current ring of Earth due to high-energy protons. The AGILE (Advanced enerGetic Ion eLectron tElescope) instrument is developed as a new-generation space instrument to provide a compact, low-mass, low-power, and low-cost particle detector. AGILE aims at identifying ions (H to Fe) and electrons and measuring their energy in a wide range of 1–100 MeV per nucleon and 1–10 MeV, respectively. AGILE will be the first space-particle detector to perform real-time discrimination using the Pulse Shape Discrimination (PSD) technique [4] .

The AGILE instrument will aim at understanding Solar Energetic Particles (SEPs) and Anomalous Cosmic Rays (ACRs). SEPs are not a fully understood phenomena [5], and understanding the precise contribution of coronal mass ejections has not been determined. Moreover, AGILE will be able to discriminate isotopes, allowing us to understand the dynamics in SEPs where $^3$He-rich events have been previously detected. Figure 1 shows such a case of an SEPs event. AGILE's energy range is also sensitive to ACRs that are produced through the interstellar neutral gas which is ionized. The study of ACRs and their trapping in the magnetosphere are of significant importance in the study of heliospheric charged particle dynamics [6,7]. Accurate measurements of the ACRs' composition and

fluxes in the outer belts by the AGILE instrument can result in a deeper comprehension of the ACRs' dynamics within the solar system, the general properties of the heliosphere, and the nature of interstellar material [8].

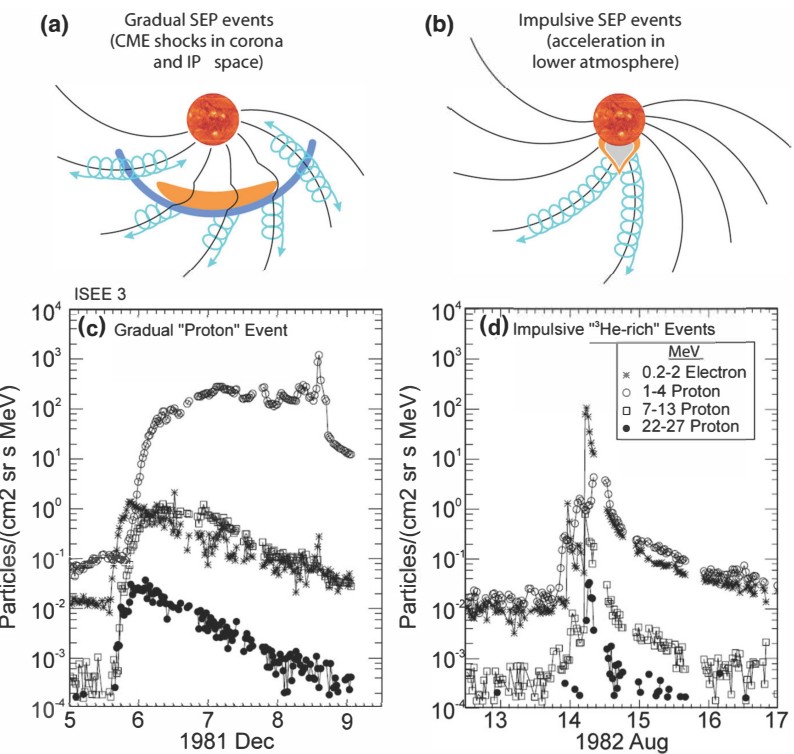

**Figure 1.** Gradual and impulsive SEP events, adapted from [5,9]. (**a**)—gradual events produced by the acceleration driven by coronal mass injection (CME) shock wave, (**b**)—impulsive events produced by a solar flare. (**c**,**d**)—flux profiles of different particles (protons and electrons) for the two types of SEP events discussed.

For space experiments, one of the most used discrimination techniques is $\Delta E - E$. This method uses the difference in energy deposition in at least two consecutively stacked detector elements for various particle species. It requires a particle to pass through the first element and stop in the second one. However, the use of multiple detectors makes the detection system more complex. It also increases the instrument electronic noise due to requiring non-independent channels for these layers. The Pulse Shape Discrimination technique [10,11], on the other hand, can be applied to a single detector and, thus, each detectors' channels can be isolated. In particular, it is known that a single solid-state detector can be used for PSD if a particle loses all of its energy within the detector medium [12]. However, in order to fully exploit the PSD method, very fast detection systems are required (detectors, front-end electronics, and samplers), since the typical signal duration in a Si detector with a thickness of a few hundreds of microns is of the order of tens of nanoseconds [13].

The first prototype of AGILE is planned for launch in 2022 on board a 6U CubeSat [14], designed by the Genesis Engineering company [15]. The first prototype will focus on proving the capability of PSD methods and the instrument design to perform discrimination of low-energy particles in space.

## 2. Methods and Materials

The AGILE detector is composed of solid-state Si layers (300 μm) manufactured by Micron Semiconductor LTD with a circular, 20-mm diameter active area. The current flying prototype will include three of such layers, which will cover partially the global range of 1–100 MeV/n. The final version of AGILE will include more layers in order to cover the range of discrimination up to 100 MeV/n for H-Fe ions. The amplitude of signals

produced by energy deposition in the silicon medium is highly dependent on the particle types and their initial energies. Two chains of amplification were implemented in the read-out electronics to cope with the considerable amplitude range. Indeed, this 2-channel amplification allows for insuring the signal quality over 4 orders of magnitude, from $10^{-6}$ A to $10^{-2}$ A. The dual-gain amplification read-out chain (or Front End Electronics, FEE), designed by the AGILE collaboration (see Figure 2), was adapted from a previous design developed for high-energy physics applications [16]. It uses low power consumption SMD components to provide each layer of detection with two independent output channels: referred to as "low-gain" and "high-gain", respectively. A 50-gain ratio is implemented between the low-gain and the high-gain channels. These components were selected to react quickly to particle signals, allowing their pulse shape analysis. This is important for the particle identification methods (Section 3) where charge collection times are in the order of tens of nanoseconds.

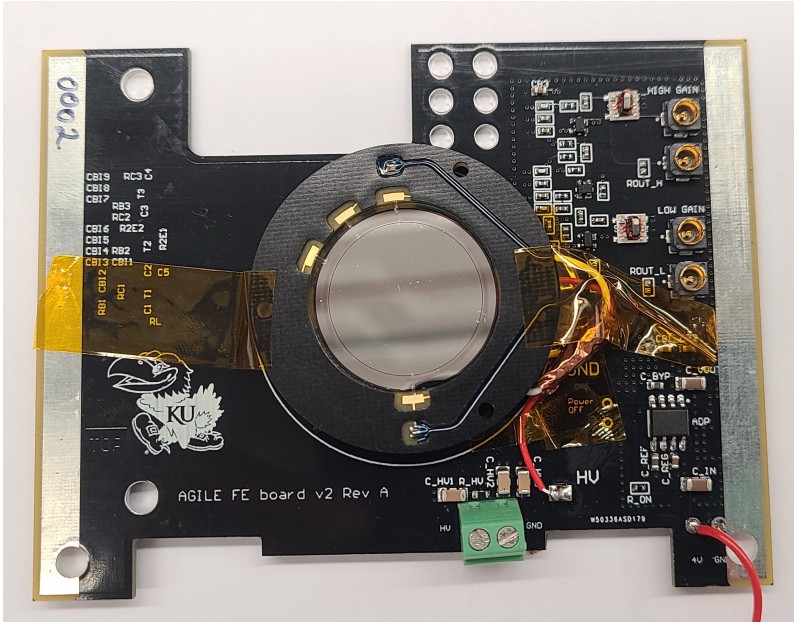

**Figure 2.** Photograph of the Micron Semiconductor Ltd MSD020 Si sensor connected to the read-out electronics board (9 cm by 8 cm).

The range of the output voltage of the amplifier chains corresponds to the operational range of the digitizing chip used in AGILE (1 mV to 1000 mV). The PSEC4 sampler chip [17] chosen for the AGILE prototype provides a fast, robust, and low-power waveform recorder with a 6-channel input. PSEC4 has a 256-sample-deep switch-capacitor array (SCA) with an adjustable sampling rate (5–15) GSa/s, which allows AGILE to fully digitize the rising edge (∼10 ns) of the Si detector signals. PSEC4 is controlled by the instrument microcontroller Teensy 4.1 and an embarked dedicated small FPGA used for fast trigger handling. The Teensy 4.1 [18] is equipped by an ARM Cortex-M7, providing AGILE with a 600-MHz core used for the control of the instrument's different units (power monitoring, sensors activation, and on-board processing of raw data) and for handling all the communications with spacecrafts. In addition to previous characteristics, all previous devices in AGILE were chosen to accommodate the restricted form factor and power consumption imposed by the space requirement to fit in a CubeSat. The whole instrument fits in 10-cm cube with a maximum power consumption of ∼1.8 W.

AGILE's development relies on the full simulation of the instruments' electronics, from the interaction of the particles in the Si detectors and the tracking of the induced charges created in the detector electric field to the two-gains output of the FEE board. In order to ascertain the capacity of AGILE to discriminate particle species and to measure their energies, the Pulse Shape Discrimination method was developed on the basis of the

simulation results. The simulation serves to create a database and tables that can be used for on-board live discrimination, using the Teensy microcontroller's processing capacity. This would allow for a reduction of the amount of data required to download to Earth, which is an important factor for space missions due to the limit of downlink (less than 1 megabyte per day predicted for the AGILE prototype mission). First, the energy deposition profiles in each layer of the Si detector are simulated using GEANT4 software [19]. Then, we simulate the detector response (output signal: I(t)) by passing Geant4 energy deposition profiles to Weightfield2 software [20], a simulation tool for silicon and diamond detectors that was adapted to AGILE. Finally, the read-out electronics (amplifier output: V(t)) response is simulated by passing the Weightfield2 output to the circuit simulator LTspice [21]. Figure 3 summarizes the simulations and illustrates the outputs at each of the stages.

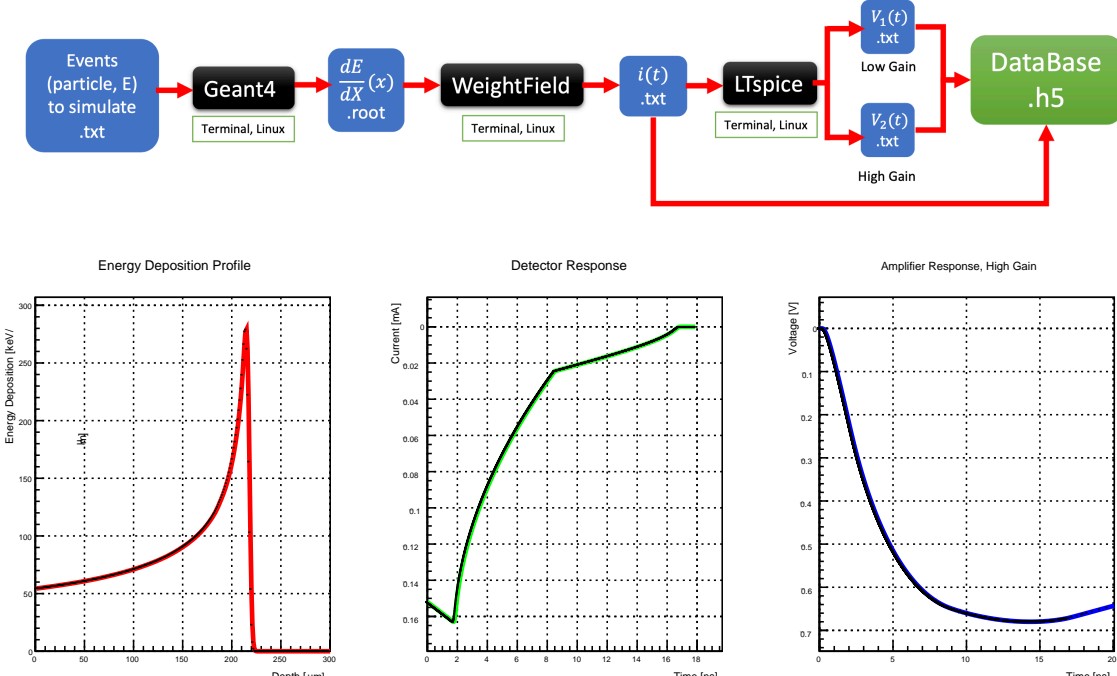

**Figure 3.** Simulation chain and simulation output examples; from left to right: energy deposition profile simulated in GEANT4 ($\frac{dE}{dx}(x)$); detector response (current pulse (i(t)) in Weightfield2; amplifier (high-gain) output (voltage pulse V (t)) in LTspice.

The simulation uses a random distribution of incident particles: ion type (H-Fe) and energy (1 MeV/n–100 MeV/n) were randomly generated using a uniform distribution and then passed to the simulation chain described above. Full details of the simulation can be found in [4]. These simulations are to be fitted to the results of the tests that are performed in lab and at beam test facilities.

Initial lab tests were performed at the University of Kansas, focusing on the Si detectors and read-out electronics. In the setup, 20-mm diameter p-type MSD020 Si detectors (with thicknesses of 300 μm) were each connected to one of the front-end electronics boards and enclosed in an aluminum mock-up of the AGILE CubeSat payload (Figure 4). Tests were performed in different configurations from only one active FEE board to all three FEE boards active in order to understand cross-interactions between layers. A Keithley 2410 SourceMeter was used to provide bias voltage (100 V) to the Si sensor. To power the FEE board, a 5-V DC voltage was provided by a USB power supply. The tests were conducted with the use of an alpha radioactive source: $^{241}$Am ($\alpha$-particles with energies 5.486 MeV (85%), 5.443 MeV (13%), and 5.388 MeV (2%)). The tests were performed at room temperature.

The signals from FEE were digitized using Teledyne LeCroy WaveRunner 640Zi oscilloscope with two input channels, low- and high-gain, respectively. The signals were

sampled at 13 GSa/s over a 300-ns window of acquisition. The trigger threshold was set as 2 mV for the acquisition. The oscilloscope's full 4-GHz bandwidth was used during signal collection.

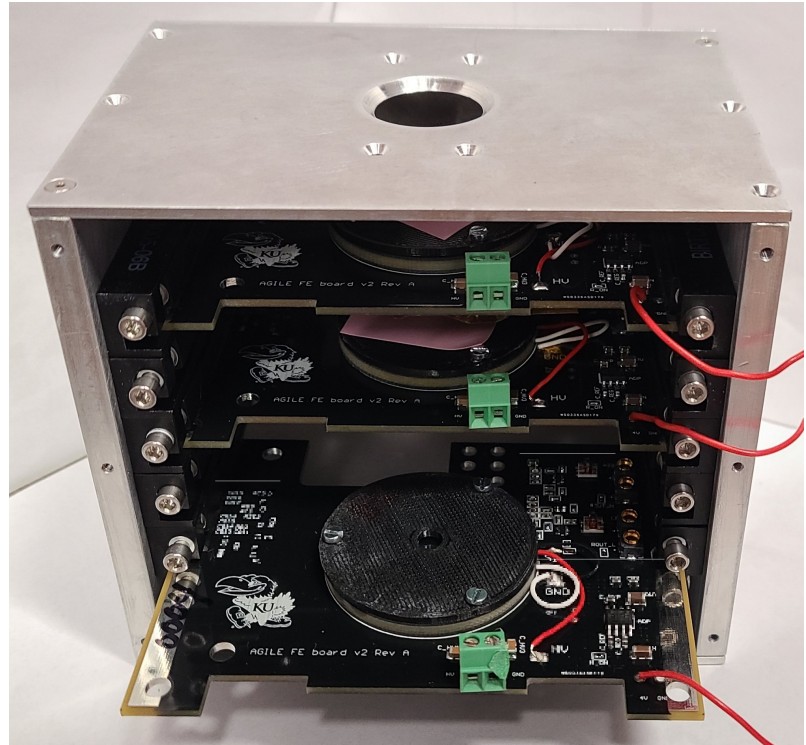

**Figure 4.** A photograph of the CubeSat mock-up used in the laboratory tests.

A photograph in Figure 4 shows the mock-up of the AGILE payload used during the laboratory tests during a 3-layer test.

## 3. Results

Simulations allowed the development of a specific identification method for AGILE by applying Pulse Shape Discrimination on the signal output of each layer. Particles deposit different amounts of energy along their path in silicon following the Bethe–Bloch curve [22], usually described by the energy loss $E$ per unit path length $x$, $\frac{dE}{dx}(x)$. This creates difference in the shape of the output signal. The development focused on only using amplitude and time measurement on the signal to characterize the interacting particle (ion type and its energy). The method retained for the AGILE prototype starts by identifying the layer where the particle stops by using a specific trigger, and retrieving this signal. Then, the key features of this signal are extracted, namely, the maximum amplitude and a timing value. The timing values can be measured at different points of the pulse on the rising edge (rise times) or decreasing edge (decay times). These values can be compared to tables created from simulations to identify the hitting particle and its initial energy. This global method is summarized in Figure 5 [4].

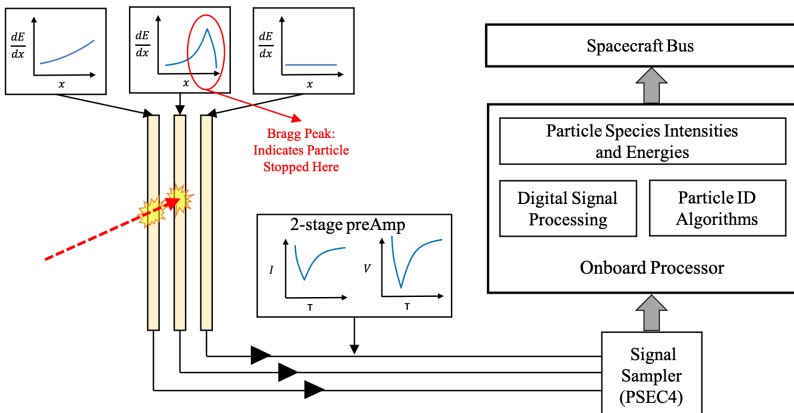

**Figure 5.** Simplified schematics of AGILE's processing of signals produced by interacting particles. In the represented case, the particles stop in the second layer.

The simulations showed that by relying only on the amplitude and time characteristics of the signal, the particle identification was possible. Figure 6 shows how these two values allow a clear distinction of the ion type. In the Figure 6, the maximum amplitude and the 85% decay time (the time for the amplitude to decrease to 85% of the maximum amplitude) are used. This plot can be turned into a table which AGILE can embark for on-board identification. The signal amplitude is directly proportional to the particle energy and is used alone for the energy measurement, as seen in Figure 7.

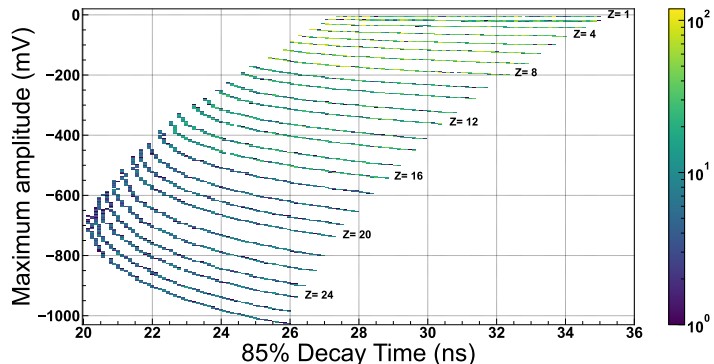

**Figure 6.** Maximum amplitude vs. 85% Decay time for H-Fe ions stopping in the detector for low gain. The color bar shows the number of events in each bin of this 2D histogram.

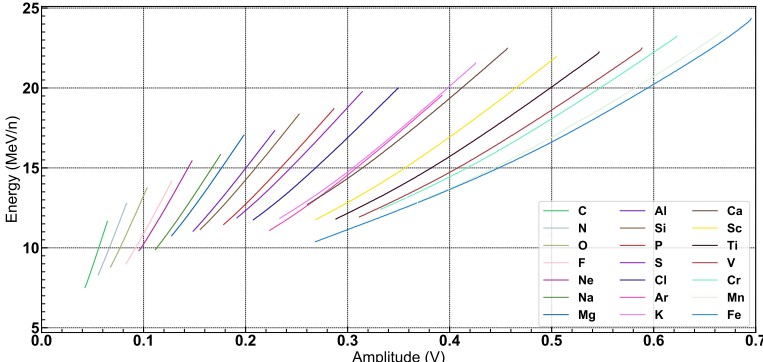

**Figure 7.** Estimated energy as a function of the measured maximum amplitude in layer 1 when the determination of particle ID is possible. Lines from left to right are for C to Fe. Figure extracted from [4].

The analysis of this simulation provides the assertion of a robust discrimination in the range of operation for the stopping particles while excluding the signals of non-stopping particles. The measurement of energy from only one layer shows promising resolution, even when accounting for electronic noise (∼5% from [4]).

Simulations also showed the global linear effect of temperature on AGILE's detector. By using the simulations and on-board temperature measurements, the AGILE instrument can correct the timing and amplitude values. However, the same simulations showed how the field of view could limit the particles' discrimination for the AGILE prototype. Indeed, particles impacting the detector with a high angle are seen as depositing more energy in the layers and, thus, broaden the identification line of Figure 6, leading to possible overlaps. AGILE is, therefore, limited in its prototype version to a field of view of 50° [4].

The simulations' work is to be confirmed by full-extent laboratory tests and a beam test campaign. However, Figure 8 shows the promising results of initial laboratory tests through the record of alpha particles emitted by an $^{241}$Am source when three silicon layers are active (alpha particles stopped in the first layer).

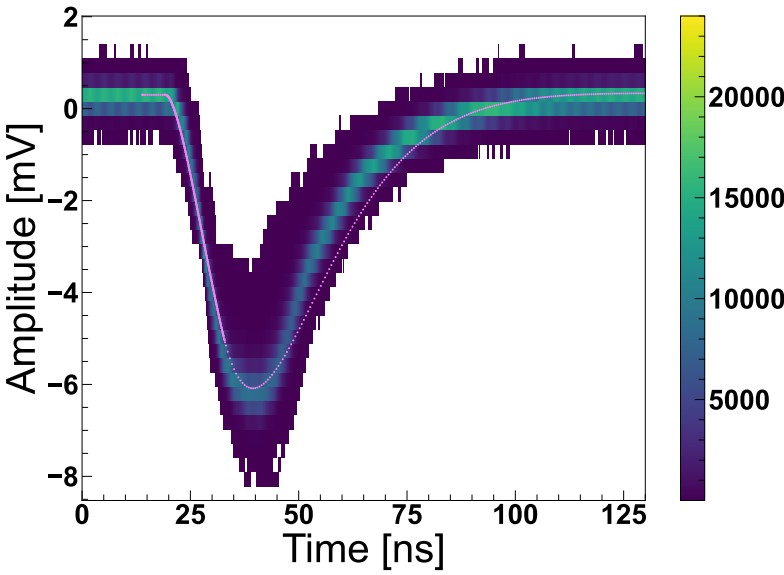

**Figure 8.** Two-dimensional density histogram of recorded signals for the high-gain channel on the first layer of AGILE. The simulated signal is plotted as a dotted violet line.

While limited to only one particle, this test showed confirmation of the performance of the AGILE hardware. The global uncertainty on the key features extracted for these collected signals showed a dispersion close to the simulation's value. The total noise measured during the laboratory test showed levels less than 1 mV for low gain and ∼10 mV for high gain.

## 4. Conclusions

AGILE will occupy 1 unit of the GENSAT-1 CubeSat with the flight scheduled for 2022 (Figure 9). This initial version of the telescope will employ a stack of three layers of fast Si detectors to provide the first use of real-time Pulse Shape Discrimination techniques in space. To achieve the required identification efficiency, the detection layers are monitored by custom-made read-out electronics, and performance-matched sampling devices. The identification of ions (H-Fe) and their energy measurements can be accomplished by using Pulse Shape Discrimination directly in space. AGILE will use a novel method to precisely identify charged particles. Rise time and maximum amplitude provides excellent identifiers of a particle's charge and mass. Energy can be estimated directly from the maximum amplitude of this same signal. The main advantage of this new method, with respect to the widely used $\Delta E - E$ technique, is in the use of both amplitude and timing information of

the signal and the possibility of particle discrimination using only the detector layer where the particle stops. AGILE's development showed the difficulty of handling a broad range of ions and developed an adapted read-out board with two specific amplification channels for light and heavy ions. The prototype version of AGILE will not cover the full range of 1–100 MeV/n, but will stop oxygen up to ∼22 MeV/n or iron up to ∼45 MeV/n with three thin layers of silicon. Based on the simulations and initial lab results, the AGILE instriment can provide a robust discrimination of a large variety of ions (from H to Fe) and an energy estimation with a resolution of less than ∼5% and an ID discrimination efficiency up to 100%, within the range of its energy acceptance.

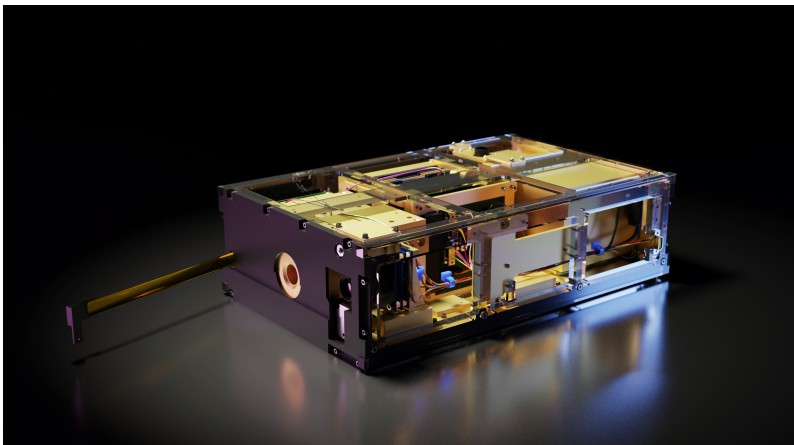

**Figure 9.** GENSAT-1 CubeSat rendered model designed by the Genesis Engineering company [15] that will carry the AGILE instrument in space.

Broader calibration will be performed in a test beam campaign at Brookhaven National Laboratory in early 2022 to prepare the instrument for space flight acquisition.

**Funding:** This work is supported by the Heliophysics Technology and Instrument Development for Science (HTIDS) ITD and the LNAPP program part of the NASA Research Announcement (NRA) NNH18ZDA001N-HTIDS for Research Opportunities in Space and Earth Science—2018 (ROSES-2018).

**Data Availability Statement:** The data presented in this study are available on request from the corresponding author. The data are not publicly available due to ongoing instrument development.

**Acknowledgments:** The author thanks the other members working at the University of Kansas and the NASA Goddard Space Flight Center for their contributions to the development of AGILE. The author thanks Tom Flatley and the GenSat-1 team for providing a CubeSat platform and engineering support. The author would like to thank the organizing committee of the 2021 WORKSHOP ON PICO-SECOND TIMING DETECTORS FOR PHYSICS.

**Conflicts of Interest:** The author declares no conflict of interest. The funders had no role in the design of the study; in the collection, analyses, or interpretation of data; in the writing of the manuscript, or in the decision to publish the results.

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
