# Peer review of "AGILE: Development of a Compact, Low-Power, Low-Cost, and On-Board Detector for Ion Identification and Energy Measurement"

_instruments, doi:10.3390/instruments6010016_

Round 1
Reviewer 1 Report
Minor Changes:
- Please go over the text for simple typos such as page 2, line 47 "Puslse", or line 89 give space between "10cm"
- Please use "MeV" everywhere consistently. Lines 62 and 63 both have "Mev" - line 74, please use "tens of nanoseconds". current version might cause some confusion. - In the methods section, please mention what the individual gain values for Low-Gain and High-Gain amplifications. - Line 72, you should give some values for the definition of "fast" - Figure 2: I am not sure if this figure actually provides any information for the reader. A schematics of the readout chain would actually help. - On the methods section, please provide information on the noise level of your amplification system. - Line 91, please give the definition of "deep" simulation. What makes it different than a regular Geant4 simulation? - Please provide the physics package used in Geant4 simulations. - Figure 6, x-axis: what do you mean by decay time? rise time (or fall time, since your signal is negative)? the term "decay time" can be misleading, since in the text you use rise time. - I assume you used 3 layers of Si detectors on the tests. This should be clearly stated on line 115. - Please provide some information on the activity level of your Am-241 source. - Figure 6 caption mentions "H-Fe ions". Are these Iron ions? if so please make it clear for the naive reader. - It is not clear what new information Figure 9 provides to the reader. I think it can be removed.
Major Changes:
- You mention both in the introduction and the conclusion that the device will use DelE - E technique. However, this was not applied to your simulation or test data. It would improve the paper if you can show that this approach will work for your system. The radioactive source tests need to be explained/reported better: - In line 111 you mention that the test results and simulations are fitted. You should report such fit or comparison. For example, in Figures 6 and 7 you report the simulations, it would be great to see the comparison of real test and simulation data for the alpha particles. You should provide the same max Ampl. vs Decay Time and Energy vs Amplitude plots for this data/simulation comparison on alpha particles. - The paper claims that Pulse Shape Discrimination is the method that is used, however, all the analysis is based on amplitude and rise time. Pulse shape discrimination means detecting the change of signal within a small "gate" time window. Figure 5, and the text, clearly mention that the amplitudes in 3 layers are compared. The Bragg peak amplitude is the main criteria here. - Line 159 mentions the acceptance angle window for the setup. This is very critical information and needs to be tested with simulations and test beam. How does the performance on identification change with the angle? Is it possible that that the well-separated lines of energy and amplitude actually start to cross each other with the angle of incidence of the particles? This is very likely, then, probably the detector window needs to be designed accordingly. - Line 160, claims that simulations and laboratory results are in agreement. However, there is no evidence given to the reader. - Figure 8 shows a simple scope signal of the Si detectors, the text (line 161) says this is very promising. This statement needs more explanation, what is the promising thing here? - Line 164, again, claims that alpha particle test results confirm the performance of the detector. For such a claim, you need to provide both simulation and the tests of the same particle (alpha) with the same energy and incidence angles. Then it would be an acceptable statement. - Line 165, talks about a global uncertainty, and dispersion. Please expand this and support your statements with data/plots.
Author Response
Minor Changes:
- Please go over the text for simple typos such as page 2, line 47 "Puslse", or line 89 give space between "10cm"
Space has been added and word changed to ‘pulse’
- Please use "MeV" everywhere consistently. Lines 62 and 63 both have "Mev"
MeV has been changed to be used everywhere in the text.
- line 74, please use "tens of nanoseconds". current version might cause some confusion.
“~10s” was changed to tens of “nanoseconds”.
- In the methods section, please mention what the individual gain values for Low-Gain and High-Gain amplifications.
The gain ratio information (50) was added in line 74.
- Line 72, you should give some values for the definition of "fast"
Current version of the AGILE read-out electronics design is not public however, the design is based on a technologies developed for picoseconds application at CERN that is the reference [16].
line 70: “The dual-gain amplification read-out chain (or Front End Electronics, FEE) designed by the AGILE collaboration, see Figure 2, was adapted from a previous design developed for high-energy physics applications [16].”
- Figure 2: I am not sure if this figure actually provides any information for the reader. A schematic of the readout chain would actually help.
The author thinks that providing a visual view of the FEE board offers the reader an idea of the compactness of the instrument. As the design is not public the detailed schematics cannot be provided and the author thinks a simplified schematics would not provide relevant information to the readers.
- On the methods section, please provide information on the noise level of your amplification system.
Global noise has been added to the results section (line 172): “The total noise measured during the laboratory test showed levels less than 1 mV for Low Gain and ∼10 mV for High Gain”
- Line 91, please give the definition of "deep" simulation. What makes it different than a regular Geant4 simulation?
The word “deep” was replaced by “full” to express the extensive simulations that include all the stages of electronics from particle interaction to signal sampling. It is not a regular Geant4 simulation as Geant4 does not allow the tracking of each electron-hole pairs in silicon detectors. Thus, the author briefly describes the extent of the simulation from the line 93 to 114. The author already pointed out the use of other software that extended the simulation from just Geant4 in this text (Weighfield2 and LTspice software). The author also refers to a previous paper detailing the simulation process “Full details of the simulation can be found in [4].”
- Please provide the physics package used in Geant4 simulations.
The author does not think that adding more details and focusing on Geant4 would benefit to the manuscript. The author however reference a previous paper focusing on the simulation and the method of discrimination “Full details of the simulation can be found in [4].”
- Gautier, F.; Greeley, A.; S.G., K.; , Isidori, T.; Legras, A.; Minafra, N.; Novikov, A.; Royon, C.; Schiller, Q. A novel technique for real-time ion identification and energy measurement for in situ space instrumentation. arXiv:2103.00613 2021.
- Figure 6, x-axis: what do you mean by decay time? rise time (or fall time, since your signal is negative)? the term "decay time" can be misleading, since in the text you use rise time.
Clarified time measurements at line 144: “The timing values can be measured at different points of the pulse on the rising edge (rise times) or decreasing edge (decay times)”
Added explanation in line 150. “In the Figure [6], the maximum amplitude and the 85\% Decay time (time for amplitude to decrease to 85\% of maximum amplitude) are used.”
Further details are in paper [4]
- I assume you used 3 layers of Si detectors on the tests. This should be clearly stated on line 115.
The author rephrased line 119 to: “Tests were performed in different configuration from only one active FEE board to all three FEE boards active in order to understand cross interaction between layers.“
The author changed the caption of Figure 8 to: “2D density histogram of recorded signals for the Low Gain channel on the first layer of AGILE. The simulated signal is plotted as a dotted violet line.”
The author mentioned in line 167: “when 3 silicon layers are active (alpha particles stopped in the first layer).”
- Please provide some information on the activity level of your Am-241 source.
The exact activity level of the source has not been measured as not of significant importance for the initial tests performed. Its value was significantly lower than the maximum detection rate of AGILE due to the absence of observed overlaps. The author deems the activity level not of importance to the paper, the main point shown being the signal shape acquisition.
- Figure 6 caption mentions "H-Fe ions". Are these Iron ions? if so please make it clear for the naive reader.
It indeed states from hydrogen ions to iron ions. Author mentions several time in the manuscript the extent of the discrimination of ions (using also the same symbols) and judges the symbol of H and Fe as rudimentary enough to be acceptable.
- It is not clear what new information Figure 9 provides to the reader. I think it can be removed.
The author use the Figure 9 to provide the reader an idea of the structural environment in which AGILE prototype will fit. As the reader is not always familiar with a CubeSat satellite, the author thinks it is appropriate to keep this figure.
Major Changes:
- You mention both in the introduction and the conclusion that the device will use DelE - E technique. However, this was not applied to your simulation or test data. It would improve the paper if you can show that this approach will work for your system
The DE-E method is not used in this paper. This method is presented at line 42 in the introduction as the usual and older method used in space. AGILE proposes to adapt the PSD method to space instrument for the first time and not to reproduce DE-E method.
The DE-E method is again mentioned in the conclusion at line 185 for the reader to understand the difference of AGILE method (PSD) compared to older instruments that used the DE-E method.
The radioactive source tests need to be explained/reported better:
- In line 111 you mention that the test results and simulations are fitted. You should report such fit or comparison. For example, in Figures 6 and 7 you report the simulations, it would be great to see the comparison of real test and simulation data for the alpha particles. You should provide the same max Ampl. vs Decay Time and Energy vs Amplitude plots for this data/simulation comparison on alpha particles.
The simulated value for alpha was added to the figure 8.
- The paper claims that Pulse Shape Discrimination is the method that is used, however, all the analysis is based on amplitude and rise time. Pulse shape discrimination means detecting the change of signal within a small "gate" time window. Figure 5, and the text, clearly mention that the amplitudes in 3 layers are compared. The Bragg peak amplitude is the main criteria here.
When the author speaks of pulse shape discrimination, he refers to the one defined in the reference [10] and more recently in the application paper [11].
“When an ionizing particle enters a semiconductor radiation detector a voltage pulse is produced. The final pulse height is determined by the particle energy, while the range of the particle affects the shape of the pulse. The voltage pulse therefore contains information about the particle type. The pulse height at a fixed time during the pulse rise time is used as the range dependent quantity.”
In order to perform the PSD method the pulse need to be reconstructed. From this, specific features (amplitude and time, decay time in AGILE case) are extracted that are characteristics to mass, charge and energy of the stopping particle. The exact features chosen for AGILE are described in [4]. The amplitude alone is not sufficient to describe completely the particle and its energy in one layer only. This is the main difference with DE-E method.
- Line 159 mentions the acceptance angle window for the setup. This is very critical information and needs to be tested with simulations and test beam. How does the performance on identification change with the angle? Is it possible that that the well-separated lines of energy and amplitude actually start to cross each other with the angle of incidence of the particles? This is very likely, then, probably the detector window needs to be designed accordingly.
Line 160 to 164:
“However, the same simulations showed how the field of view limits the particles discrimination for AGILE prototype. Indeed, particles impacting the detector with a high angle are seen as depositing more energy in the layers and thus broaden the identification line of Figure 6. AGILE is thus limited in its prototype version to a field of view of 50°”
What reviewer mentions is correct and was already stated in the manuscript in the lines cited above. The author revised the text to make it clearer without repeating analysis performed in Reference [4]. (Section 4.2.1, paragraph Incident angle variation figure 17).
Concerning laboratory tests and beam tests of angles, this is out of scope of this proceeding but will be addressed by the AGILE team in future publications.
- Line 160, claims that simulations and laboratory results are in agreement. However, there is no evidence given to the reader.
The author changed the sentence as it could be misleading compared to sentence line 115 where the tests described were presented as “initial lab tests” and not full extent tests.
line 165: “The simulations work is to be confirmed by full extent laboratory tests and beam test campaign.”
See further response at the end
- Figure 8 shows a simple scope signal of the Si detectors, the text (line 161) says this is very promising. This statement needs more explanation, what is the promising thing here?
Figure changed. See further response at the end
- Line 164, again, claims that alpha particle test results confirm the performance of the detector. For such a claim, you need to provide both simulation and the tests of the same particle (alpha) with the same energy and incidence angles. Then it would be an acceptable statement.
See response at the end
- Line 165, talks about a global uncertainty, and dispersion. Please expand this and support your statements with data/plots.
The author added the simulation pulse expected for the alpha particle to the laboratory measurements in figure 8. This figure shows detection and measurement of the alpha particles by the AGILE instrument. This shows the capacity of the design presented here to retrieve the pulse shape for the discrimination method presented in reference [4]. Values of amplitude and noise are in the order of the expected values presented in that same reference. These results are not a full extent test of the AGILE instrument.
The author rephrased the paragraph to show that the initial test results presented are not full test of the AGILE instrument.
The author would like to point out the status of this manuscript as a proceeding for a talk given at the WORKSHOP ON PICO-SECOND TIMING DETECTORS FOR PHYSICS and thus the manuscript focus on what was presented at workshop.
The presentation of full extent tests results of AGILE in laboratory and at a beam test facility are planned for other publications.
Reviewer 2 Report
The paper is well written (even though there are several typos and a couple of sentence I’m not sure are correct in english) and clear. I recommend for publication essentially as its is, after the needed corrections.
Here’s the list of comments:
MA-1) L28: fist
OP-1) Fig.1: “C” and “D” would be better as “c” and “d” or “(c)” and “(d)”. Isn’t it? This is really a very minor comment, just aesthetic
MI-1) L46: why the use of multiple detectors increases the instrument noise? The only “mechanism” I can imagine is if keep the total thickness of the system. Maybe, most likely, I’m wrong (I’m not an expert of DeltaE-E, even being quite expert of silicon detectors), but maybe is better to elaborate a little bit more the concept, to let the reader understand more easily.
MA-2) L47: Puslse
MA-3) L60: mm^2? Reading later I understand that maybe is the diameter, but here you talk about “area”. Clarify.
MA-4) L62 and also L63: MeV
MI-2) Fig.3: is possible to have a figure so large?
MA-5) L93: field,up. The space after the “,”” is missing
MI-3) L156: “when accepting important field of view for AGILE”. I understand what you mean, also by reading the next sentence, but I’m not sure if this sentence is correct in english
MA-6) L164-145: “these” “signal”
MI-4) L182-183: “stop oxygen up to … and iron up to …”. I’m not sure this is correct in English. “will stop AT oxygen”, maybe?
(Legend: MI = Minor, MA = Mandatory, OP = Optional)
Author Response
- MA-1) L28: fist
Replaced by First
- OP-1) Fig.1: “C” and “D” would be better as “c” and “d” or “(c)” and “(d)”. Isn’t it? This is really a very minor comment, just aesthetic
The figure was modified with (a) (b) (c) & (d)
- MI-1) L46: why the use of multiple detectors increases the instrument noise? The only “mechanism” I can imagine is if keep the total thickness of the system. Maybe, most likely, I’m wrong (I’m not an expert of DeltaE-E, even being quite expert of silicon detectors), but maybe is better to elaborate a little bit more the concept, to let the reader understand more easily.
Due to requiring at least two layers of detectors, DeltaE-E method cannot completely isolate the channels of each detectors creating more opportunities of interferences and noise. The PSD relies on only one layer so when using several layers (for higher energetic particles) each channel can be isolated more easily.
Added line 46: “It also increases the instrument electronic noise due to requiring non-independent channels for these layers”
- MA-2) L47: Puslse
Changed to “pulse”
- MA-3) L60: mm^2? Reading later I understand that maybe is the diameter, but here you talk about “area”. Clarify.
It was indeed an error and should be the diameter. The sentence was rephrased
- MA-4) L62 and also L63: MeV
The author checked that MeV was used throughout the manuscript
- MI-2) Fig.3: is possible to have a figure so large?
The author used the latex template with the \widefigure option provided.
- MA-5) L93: field,up. The space after the “,”” is missing
The space has been added
- MI-3) L156: “when accepting important field of view for AGILE”. I understand what you mean, also by reading the next sentence, but I’m not sure if this sentence is correct in english
The sentence has been rephrased to “However, the same simulations showed how the field of view could limit the particles discrimination for AGILE prototype.”
- MA-6) L164-145: “these” “signal”
This has been corrected
- MI-4) L182-183: “stop oxygen up to … and iron up to …”. I’m not sure this is correct in English. “will stop AT oxygen”, maybe?
The meaning here is the range of stopped particle 1 to 45 MeV/n for iron and 1 to 22 MeV/n for oxygen. No change was performed
(Legend: MI = Minor, MA = Mandatory, OP = Optional)
Reviewer 3 Report
The paper "AGILE: DEVELOPMENT OF A COMPACT, LOW POWER, LOW COST AND ON-BOARD DETECTOR FOR ION IDENTIFICATION AND ENERGY MEASUREMENT" presents a nice study for a prototype/proof of concept detector that will be installed on a satellite.
We do not find methodological errors in the paper, although the studies on how to discriminate elements are not fully complete yet.
A list of minor corrections/typos:
-Figure 1: (C) and (D) should be small letters, (c) and (d) with parenthesis
-Caption of Figure 1: Put (a), (b), (c), (d) with parenthesis
- l.60 : are of 20 mm-> Not clear, area should be in mm^2 (a space before mm is also needed)
-l. 62 and 63: "Mev" -> "MeV"
l.72: "to not lose" -> "not to lose" ?
l.97: referece to "Teensy"
l.133: rephrase , or example: "path in silicon, following the Beth-Block curve (ref) describing the energy loss per..."
Author Response
The paper "AGILE: DEVELOPMENT OF A COMPACT, LOW POWER, LOW COST AND ON-BOARD DETECTOR FOR ION IDENTIFICATION AND ENERGY MEASUREMENT" presents a nice study for a prototype/proof of concept detector that will be installed on a satellite.
We do not find methodological errors in the paper, although the studies on how to discriminate elements are not fully complete yet.
A list of minor corrections/typos:
- Figure 1: (C) and (D) should be small letters, (c) and (d) with parenthesis. Caption of Figure 1: Put (a), (b), (c), (d) with parenthesis
The figure was modified with (a) (b) (c) & (d)
- 60 : are of 20 mm-> Not clear, area should be in mm^2 (a space before mm is also needed)
It was indeed an error and should be the diameter. The sentence was rephrased
- 62 and 63: "Mev" -> "MeV"
The author checked that MeV was used throughout the manuscript
- 72: "to not lose" -> "not to lose" ?
The sentence was rephrased to: “These components were selected to react fast to particles signals allowing their pulse shape analysis.”
- 97: referece to "Teensy"
Line 85 describes AGILE microcontroller: Teensy. Replaced “the Teensy” by “the Teensy microcontroller”
l.133: rephrase , or example: "path in silicon, following the Beth-Block curve (ref) describing the energy loss per..."
The sentence was rephrased to: “Particles deposit different amounts of energy along their path in silicon following the Bethe-Bloch curve [22], usually described by the energy loss E per unit path length x ”